# Analysis of Thermoelectric Energy Harvesting with Graphene Aerogel-Supported Form-Stable Phase Change Materials

**DOI:** 10.3390/nano11092192

**Published:** 2021-08-26

**Authors:** Chengbin Yu, Young Seok Song

**Affiliations:** 1Department of Materials Science and Engineering, Research Institute of Advanced Materials (RIAM), Seoul National University, Seoul 08826, Korea; ycb0107@snu.ac.kr; 2Department of Fiber Convergence Materials Engineering, Dankook University, Yongin-si 16890, Gyeonggi-do, Korea

**Keywords:** phase change material, Seebeck effect, thermoelectric energy harvesting

## Abstract

Graphene aerogel-supported phase change material (PCM) composites sustain the initial solid state without any leakage problem when they are melted. The high portion of pure PCM in the composite can absorb or release a relatively large amount of heat during heating and cooling. In this study, these form-stable PCM composites were used to construct a thermoelectric power generator for collecting electrical energy under the external temperature change. The Seebeck effect and the temperature difference between the two sides of the thermal device were applied for thermoelectric energy harvesting. Two different PCM composites were used to collect the thermoelectric energy harvesting due to the different phase transition field in the heating and cooling processes. The graphene nano-platelet (GNP) filler was embedded to increase the thermal conductivities of PCM composites. Maximum output current was investigated by utilizing these two PCM composites with different GNP filler ratios. The thermoelectric energy harvesting efficiencies during heating and cooling were 62.26% and 39.96%, respectively. In addition, a finite element method (FEM) numerical analysis was conducted to model the output profiles.

## 1. Introduction

Renewable energy has been widely utilized in various areas to replace fossil fuels due to the serious energy crisis and high environmental pollution [1,2]. The development of renewable energy with high efficiency is important for general life [3,4]. Hence, solar heat energy is considered as a wonderful type of clean energy, due to easy collection and low costs [5,6]. It is necessary to collect and use external clean energy. Therefore, thermoelectric energy conversion has attracted a lot of attention due to its appropriate efficiency [7,8]. Carbon-based materials, such as carbon nanotubes (CNTs) and graphene, can absorb solar energy sufficiently and it is easy to transfer their thermal energy due to the appropriate thermal conductivities [9,10]. Some matrices with high thermal energy storage (TES) are needed to combine the carbon-based materials for energy harvesting applications. As a result, phase change materials (PCMs) are selected as working materials in TES systems [11,12]. PCMs are divided into metal, inorganic and organic materials [13,14]. Depending on the type of PCM, energy storage can be described as solid–solid, solid–liquid, solid–gas and liquid–gas upon phase transitions [15,16]. However, practical production of energy and resources requires inorganic and organic PCMs with high heat of fusion. In particular, organic PCMs exhibit high form stability and avoid super-cooling upon melting and cooling. Therefore, they are regarded as the most appropriate working material for TES applications [17,18]. Most of organic PCMs uses solid–liquid phase transitions to store the thermal energy. In other words, PCMs are able to absorb or release a large amount of heat during the phase transition process, which has high latent heat to transfer thermal energy [19,20]. The thermoelectric power generator has been utilized to resolve energy depletion and PCMs are connected to the power generator device due to their excellent thermal stability during the heating and cooling processes [21,22]. In this respect, PCMs are employed as working materials to control the degree of thermoelectric energy conversion [23,24]. The positive negative thermoelectric generator (PN TEG), which is based on N and P type semiconductors, has been utilized in the energy harvesting field using the Seebeck effect [25,26].

However, some problems of pure PCMs, such as leakage, shrinkage and liquid state, restrict their general utilizations [27,28]. To prevent PCM leakage, supporting materials are employed in the PCM composites [29,30]. The core-shell-like structure is suggested to sustain the initial solid state without any leakage during melting [31,32]. Polystyrene (PS), polyaniline (PANI) and melamine-formaldehyde (MF) have been selected as supporting shell materials to encapsulate the PCM microcapsule [33,34]. Although the microencapsulated PCM composites exhibit form stability during the phase transition process, a large portion of the working material is replaced with the supporting material. That is why the total latent heat of PCM composites decreases significantly [35,36]. Thus, three-dimensional porous aerogel is regarded as an advanced supporting material which can hold plenty of pure PCMs in the internal space [37,38]. The aerogel-supported PCM composite is fabricated by the vacuum impregnation method and the molten pure PCM is fully infiltrated into the aerogel internal space. Graphene and silica aerogels are selected as the supporting materials to manufacture the form-stable PCM composites [39].

The thermoelectric energy harvesting efficiency depends on the thermal conductivity of the PCM composite and graphene aerogel is a suitable candidate to fabricate form-stable PCM composites [40]. To enhance both the thermal conductivity and mechanical property of the graphene aerogel, graphene nano-platelet (GNP) is selected as a filler to construct a three-dimensional hybrid porous skeleton. Graphene aerogel is obtained by the reduction of graphene oxide (GO) aerogel and GO aerogel is produced by the freeze-drying method, which evaporates the whole solvent of the GO aqueous solution [41,42]. According to the mass ratio between GO and GNP, reduced-graphene oxide aerogels (rGO/GNP structures)-supported PCM composites exhibit higher thermal conductivities. PN TEGs are utilized as thermoelectric devices and two different PCM composites are connected to each side of PN TEGs [43]. PN TEGs possess a hot side and a cold side due to the internal structure of the N and P type semiconductors [44,45]. The electrons doped in the N type semiconductor start the excitation while absorbing the external thermal energy and move to the P type semiconductor [46,47]. The major carrier of P type semiconductor is a hole due to the lack of electrons and the electrons can combine with the holes. As a result, the electrons aggregate on the hot side of the PN TEG and generate a potential to promote electron movement in the circuit. Different PCM composites give rise to the temperature difference around PN TEGs during the phase transition process and the induced electrical current is generated during the change of external temperature [48,49].

In this work, polyethylene glycol (PEG) and 1-tetradecanol (1-TD) phase change materials (PCMs) were utilized to fabricate PCM composites. These two pure PCMs were infiltrated into the internal space of rGO/GNP aerogels to obtain form-stable PCM composites. rGO/GNP aerogels with different GNP ratios can influence the thermal conductivities of PCM composites and thermoelectric energy harvesting efficiencies during the phase transition process. To validate the collected electrical energy, a LED bulb was used during the heating and cooling cycles. To obtain the maximum efficiency of thermoelectric energy conversion, the PCM composites with various GNP mass ratios were investigated and some of the candidates were tested for optimization.

## 2. Experimental Sections

### 2.1. Materials

For phase change materials (PCMs), polyethylene glycol (PEG Mn = 6000) and 1-tetradecanol (1-TD) were provided by the Avention^®^ corporation and Sigma-Aldrich, respectively. Graphene nano-platelet (GNP, C grade) was selected in this study, which was purchased from XG Science Michigan, USA. In addition, Graphite powder, sulfuric acid (H_2_SO_4_), potassium permanganate (KMnO_4_), hydrochloric acid (HCl), hydrogen peroxide (H_2_O_2_) and hydrazine were utilized to synthesize graphene oxide (GO) and were purchased from Sigma-Aldrich.

### 2.2. Fabrication of PCM Composites

Graphene oxide (GO) was synthesized by using a modified Hummers’ method [50,51]. A graphite of 3 g was poured into 12 mL of sulfuric acid (H_2_SO_4_) and fully oxidized by adding 15 g of potassium permanganate (KMnO_4_). After several hours, the oxidized graphite mixture was diluted by using 700 mL of distilled water (DI) and 20 mL of hydrogen peroxide (H_2_O_2_) was added to remove the excess KMnO_4_. The purified solution was washed with 10% hydrochloric acid (HCl) and the solution was removed by using a filtering process. The oxidized graphite was diluted with DI water and neutralized by the centrifugation method at 10,000 rpm. The graphene oxide (GO) powder was obtained after applying a freeze-drying procedure. The GO powder and GNP filler were dispersed in DI water under 30 min ultra-sonication. The mass ratios between GO and GNP were 2:1, 1:1 and 1:2 to fabricate GO/GNP aerogels. These GO/GNP aqueous solutions were poured into the molds with 4 cm × 4 cm × 0.5 cm and the 3D porous aerogels were obtained by the freeze-drying method. Finally, the reduced graphene aerogels (rGO/GNP) were fabricated by using a hydrazine vapor method. The PEG and 1-TD composites were fabricated by utilizing the vacuum impregnation method, which infiltrated the liquid pure PCM into the graphene aerogel internal space. The pure PEG and 1-TD were melted at 80 °C under vacuum to remove moisture. The graphene aerogels were immersed into the liquid PCMs for several hours. Both of the PEG and 1-TD composites were obtained by using a solidification method and labelled 2:1 PEG composite, 1:1 PEG composite and 1:2 PEG composite, in accordance with the GO/GNP mass ratio. In addition, the 2:1 1-TD composite, 1:1 1-TD composite and 1:2 1-TD composite were considered.

### 2.3. Design of the Energy Harvesting System

The PEG and 1-TD composites were connected to the PN TEGs to construct the thermoelectric energy harvesting system. When the temperature was increased, the 1-TD composite started the solid–liquid phase transition due to the lower melting temperature than that of the PEG composite. Thus, the PEG composite was placed at the hot side of the PN TEG and the temperature difference was generated due to the nearly isothermal field of the 1-TD composite. However, the PEG composite was under the phase transition process, while the 1-TD composite completed the solid–liquid process, and the temperature increased rapidly at the hot atmosphere. To collect the additional electrical energy, we designed a second energy harvesting system by connecting the 1-TD composite to the hot side of the PN TEG. When the surface temperature of the 1-TD composite exceeded that of the PEG composite, the Seebeck effect acted and the induced current was harvested in the circuit. The thermoelectric energy harvesting system provided electrical energy to turn on an LED bulb. The light intensity of the LED bulb was measured and was used for the optimization of the energy harvesting system [52].

### 2.4. Characterizations

The dispersion ability of GO was analyzed by measuring the zeta potential (Zetasizer, Malvern, UK) in the aqueous solution and GO and GNP were mixed. The reduced graphene oxide (rGO) was confirmed by Fourier transform infrared spectroscopy (FT-IR, Varian660, UT, USA). The graphene aerogel porous structure was observed with the use of a Brunauer–Emmett–Teller Analyzer (BET, ASAP2010, Atlanta, USA). A field-emission scanning electron microscope (FE-SEM, Merlin compact, ZEISS, Jena, Germany) was employed to measure the surface morphologies of graphene aerogel and PCM composite under a 5 kV accelerating voltage; all of samples were coated with a Au-coater. To confirm the typical peaks of PCM composites, X-ray diffraction (XRD New D8, Bruker, Billerica, MA, USA) was utilized and 2θ from 10° to 70° was observed with a rate of 3°/min. A thermal analyzer (C-Therm TCi, C-Therm Technologies Ltd., Fredericton, NB, Canada) was utilized to measure the thermal conductivities of PCM composites. The phase transition temperature and latent heat were evaluated by using a differential scanning calorimeter (DSC4000, PerkinElmer, Waltham, MA, USA) with a nitrogen gas atmosphere. The PCM samples were heated from 15 °C to 90 °C at a speed of 10 °C/min and back to initial 15 °C upon cooling process. A potentiostat (VersaSTAT 3, AMETEK^®^ PA, USA) was utilized to measure the induced current during the change of temperature from 25 °C to 80 °C. To observe the LED light during the heating and cooling processes, a light intensity meter (LI-1400, Nebraska, USA) was used.

## 3. Numerical Analysis

To calculate the temperature profiles between the PEG and 1-TD composites, each of temperature changes was simulated by using a finite element method (FEM). The energy harvesting system consisted of the PEG composite, PN TEG and 1-TD composite. The parameters considered in this study were the density, latent heat (ΔH), heat capacity, thermal conductivities of PCM composites and phase transition temperature. A copper film was utilized to combine the PN TEGs in the thermoelectric power generator [53]. The PN TEGs were doped by Bi_2_Te_3_ and Bi_0.3_Sb_1.7_Te_3_ and the Seebeck coefficient was a function of the temperature difference during the phase transition process. The total mesh for the numerical analysis was 10,178.

The governing equation of heat transfer is as below:(1)$${\rho C}_{p}\frac{\partial T}{\partial t}+{\rho C}_{p}u\cdot \nabla T+\nabla \cdot q=Q$$
where ρ is the mass density and $C_{p}$ is the heat capacity. q is the heat transfer rate which is described as a function of thermal conductivity:(2)q=−k∇T

The specific density equation of PCMs is related to the ratio of phase 1 to phase 2. In addition, the fill-factor θ and relevant parameters ($C_{p},\;k,{\;\mathrm{and}\;\alpha }_{m}$) are defined as below:(3)$$\rho ={\theta \rho }_{\mathrm{phase}1}+\left(1-\theta \right)\rho_{\mathrm{phase}2}$$
(4)$$C_{p}=\frac{1}{\rho }\left({\theta \rho }_{\mathrm{phase}1}C_{p.\mathrm{phase}1}+\left(1-\theta \right)\rho_{\mathrm{phase}2}C_{p.\mathrm{phase}2}\right)+L\frac{\partial \alpha_{m}}{\partial T}$$
(5)$$k={\theta k}_{\mathrm{phase}1}+\left(1-\theta \right)k_{\mathrm{phase}2}$$
(6)$$\alpha_{m}=\frac{1}{2}\frac{\left(1-\theta \right)\rho_{\mathrm{phase}2}-\theta \rho_{\mathrm{phase}1}}{{\theta \rho }_{\mathrm{phase}1}+\left(1-\theta \right)\rho_{\mathrm{phase}2}}$$
where $\alpha_{m}$ is the mass coefficient during the phase transitions. The normal vector n and heat flux $q_{0}$ are functions of the heat transfer coefficient ($h_{\mathrm{air}})$ and external temperature, as presented below:(7)$$-n\cdot q=q_{0}$$
(8)$$q_{0}=h_{\mathrm{air}}\cdot \left(T_{\mathrm{ext}}-T\right)$$

When the temperature gradient is created during the heating and cooling processes, the output electrical current is calculated by using the following Seebeck coefficient, S:(9)$$S=\sqrt{\frac{k}{\sigma T}}$$
where the electrical conductivity, σ, is related to the Seebeck coefficient. Thus, the temperature difference (∆T) and the resistance of energy harvesting device (R) are defined as below:(10)$$I=\frac{S∆T}{R}$$

Finally, the efficiency of thermoelectric energy conversion is the ratio of converted energy W and the total stored energy Q is as follows:(11)$$\eta =\frac{W}{Q}$$

## 4. Results and Discussion

### 4.1. Morphology of Graphene Aerogels

The fabrication of PCM composites and thermoelectric energy harvesting efficiencies are illustrated in Figure 1. The PCM composites had high latent heat and converted their thermal energy without any leakage. The graphene aerogel was utilized as a supporting material to infiltrate pure PCM into the internal porous space. The surface tension of the graphene skeleton could restrict the PCM movement during the phase transition process. The PN TEG containing the N and P type semiconductors provided the electrical energy using the temperature difference between the two sides of the device. The induced current was proportional to the temperature difference, which acted as a key factor to increase the output electrical energy. The LED bulb could be used to examine the thermoelectric energy harvesting efficiency. The images of the graphene aerogels and zeta potentials of GO and GNP are shown in Figure 2. The GO aerogels with different GNP mass ratios exhibited brown colors (Figure 2a–c). After the hydrazine reduction treatment, all of the GO aerogels turned black, which led to the rGO/GNP aerogels as shown in Figure 2d–f. To verify the GO and GNP dispersion in the aqueous solution, the zeta potential was utilized (Figure 2g). The GO and GNP exhibited high potential values, which indicated that they were fully dispersed in the DI water [54]. The pH of the GO/GNP solution was 5.23. To confirm the reduction of graphene aerogels, FT-IR was measured as shown in Figure 3a,b. The GO typical peak was found at 3400 cm^−1^, which indicated an O-H peak. C=O and C-O peaks were obtained at 1721cm^−1^ and 1054 cm^−1^, respectively. The rGO/GNP aerogels only yielded the C=C peak around 1680 cm^−1^, which indicated that the GO functional groups were fully reduced by the hydrazine treatment. In order to confirm the structural characteristics of graphene aerogels, the Brunauer–Emmett–Teller (BET) was applied by measuring the adsorption and desorption under the isothermal nitrogen atmosphere as shown in Figure 3c,d. The surface areas of the graphene aerogels are listed in Table 1. The type III isotherms were selected to obtain the multilayer adsorption in the macro-porous structure in the high-pressure region [55,56]. All of the graphene aerogels showed high specific surface areas and the pore diameters were about 0–10 nm. This demonstrated that the graphene aerogels encompassed microporous structures. To further verify the internal structure of the graphene aerogels, the rGO/GNP aerogels were investigated by SEM analysis (Figure 3e–g). All of the graphene aerogels showed the porous structure in the range of 1–10 μm, indicating that the graphene aerogels were able to infiltrate plenty of pure PCMs effectively. These results revealed that the graphene aerogels could be selected as excellent supporting materials for manufacturing form-stable PCM composites.

### 4.2. Characterization of PCM Composites

The results of PCM composites form stability are shown in Figure 4. All the samples were placed on a hot plate with a temperature from 25 °C to 80 °C. The pure 1-TD was melted to the liquid state, while the pure PEG was kept at the solid state with a little leakage. However, all of the PEG and 1-TD composites sustained the initial solid state without any leakage, even at 80 °C. From the leakage test, the graphene aerogels-supported PCM composites were found to have excellent form stability during melting. The PEG and 1-TD weight percentages in the PCM composites are shown in Figure 5a and the results are listed in Table 2. The PEG and 1-TD composites possessed the high weight percentage of pure PCMs. In addition, the porosities of graphene aerogels are presented in Figure 5b and listed in Table 3. All of the graphene aerogels exhibited high porosities, thus indicating that these graphene aerogels could hold a large amount of pure PCM in the porous space. Figure 5c,d shows the XRD peaks of PEG and 1-TD composites, respectively. The intrinsic peaks of pure PEG were observed at 19.10° and 23.21°, which are similar to those of the PCM composites. The PCM composites had the same internal structures without significant changes. The pure 1-TD and 1-TD composites showed typical peaks around 21.32° and 24.15°. From the XRD peaks, it was inferred that there was no chemical reaction between graphene aerogels and pure PCMs. The thermal conductivities of PEG and 1-TD composites are shown in Figure 5e,f. The PEG and 1-TD composites showed the increment in the thermal conductivities from 2:1 to 1:2 GNP mass ratios [40]. The 2:1 ratio PEG composite exhibited 0.4233 W/mK and yielded an increased thermal conductivity of 0.5828 W/mK for the 1:2 ratio (the 1:1 ratio case showed 0.4929 W/mK). For the 1-TD composites, the thermal conductivities were 0.3414 W/mK, 0.4135 W/mK and 0.4974 W/mK, according to the increase in the GNP mass ratios. The 1:2 ratio PCM composites showed the best thermoelectric energy conversion efficiencies. The surface structure of pure PCM and PCM composites was analyzed by using the SEM images (Figure 6). The surface of pure PEG was smooth, while PEG composites showed wrinkle structures (Figure 6a–d) [57]. It indicated that the pure PEG was fully infiltrated into the graphene aerogels to construct form-stable PCM composites. The pure 1-TD had a layer structure and the 1-TD composites showed a different structure with respect to the GNP mass ratios (Figure 6e–h). The phase transition temperature and latent heat (∆H) were obtained by the DSC measurement, as shown in Figure 7. The results are listed in Table 4. The pure PEG started melting at 50.75 °C (T_o_) and the endset melting temperature was 69.16 °C (T_e_). The melting of pure PEG was observed at 66.81 °C and the melting enthalpy was 182.62 J/g. The PEG composites with 2:1 and 1:2 GNP mass ratios exhibited similar results in the heating and cooling cycles and a little decrease in latent heat (∆H) was observed due to the increase in GNP fillers. The pure 1-TD had lower phase change temperatures than the pure PEG and showed a melting temperature of 41.63 °C. The pure 1-TD and 1-TD composites exhibited high latent heat (∆H), which can absorb or release a large amount of thermal energy during the phase transition process. Therefore, the PEG and 1-TD composites were utilized as high latent heat thermal energy storage (LHTES) materials to construct thermoelectric energy harvesting system.

### 4.3. Thermoelectric Energy Harvesting

The temperature profiles of PEG and 1-TD composites were numerically calculated as shown in Figure 8. The 1-TD composites yielded the nearly isothermal phase transition field, while the PEG composites showed the increased surface temperature during the heating process (Figure 8a–c). The maximum temperature difference was about 16 °C in the 1st heating process and the 2nd one gave 7.5 °C (Figure 8d). After removing the heat source, the temperature of the thermoelectric energy harvesting system decreased from 80 °C to room temperature due to the air exposure. Figure 8e–g shows the cooling temperature profiles of the PEG and 1-TD composites. The 1st and 2nd maximum temperature differences calculated were 12 °C and 7 °C, respectively (Figure 8h). The comparison of experimental results with numerical calculations is shown in Figure 9. The measured currents were in good agreement with the simulation results. The LED bulb was successfully turned on by using a low voltage start-up converter (LTC 3108) during the phase transition process [58]. The 2:1 ratio PCM composites exhibited the 1st maximum LED light at 1168 s and the 2nd one was at 2569 s, during heating (Figure 9d). The thermal conductivities of PCM composites were gradually increased with an increase in the ratio.

The rate of thermoelectric energy harvesting was promoted during the phase transition process. Figure 9h shows the maximum LED brightness after removing the heat source. It was found that the time required for the test decreased with increasing the GNP mass ratio. Since the GNP filler could improve the thermal conductivities of PEG and 1-TD composites, the corresponding temperature profiles during the heating and cooling processes were analyzed (Figure 10a,b). The various combinations of PEG and 1-TD composites were examined. The temperature differences are presented in Figure 10c,d. It was found that the 2:1 ratio PCM composites might not provide the maximum thermoelectric energy harvesting efficiency. 1:1 PEG/1:2 1-TD, 1:2 PEG/1:1 1-TD and 1:2 PEG/1:2 1-TD were selected as the candidate groups by considering the large peak area of temperature difference. Figure 11a,b shows the temperature difference among the three groups. The results of currents during the heating and cooling processes are presented in Figure 11c,d. The PCM groups generated the electrical current successfully with different peak areas. To confirm the maximum output current among the PCM groups, the onset and endset times for the LED were measured during the phase transition fields, as shown in Figure 12a,b. The correlated results are listed in Table 5 and Table 6. The longest time in both the heating and cooling cycles was obtained in the 1:2 PEG/1:1 1-TD system. To further demonstrate the energy harvesting ability, the light intensity of the LED bulb was evaluated during the heating process, as presented in Figure 12c–e. The 1:2 PEG/1:1 1-TD exhibited an average intensity of 0.267 μmol m^−2^ s^−1^, which was the highest among the PCM groups. The cooling results of light intensity are shown in Figure 12f–h. The 1:1 PEG/1:2 1-TD group had a similar result to the 1:2 PEG/1:1 1-TD group, which exhibited 0.218 μmolm^−2^s^−1^. According to the light intensity results during the heating and cooling processes, the 1:2 PEG/1:1 1-TD was selected as an optimum group for energy harvesting. In order to calculate the largest current among the PCM groups, the comparison of the peak ratios was made, as shown in Figure 13a,b. The 1:2 PEG/1:1 1-TD group exhibited the highest ratios of peak area during heating and cooling, which were 1.16 and 1.22, respectively. The comparison of the numerical simulation with the experimental result is illustrated in Figure 13c,d. The maximum current was 12.45 mA at the 1st phase transition and 5.80 mA at the 2nd transition, during heating. The cooling peak showed 9.75 mA and 6.20 mA output maximum currents at the 1st and 2nd phase transitions, respectively. In addition, the thermoelectric energy harvesting efficiencies calculated during heating and cooling were 62.26% and 39.96%, respectively. These results were higher than those of the 1:2 PEG/1:2 1-TD group (i.e., 55.59% and 33.33%). The optimum group of PCM composites is expected to be applied to thermal sensing, aerospace and pyroelectric energy harvesting.

## 5. Conclusions

In this study, an advanced energy harvesting system was constructed using PCM composites. It was found that the embedded GNP led to an increase in the thermal conductivity and mechanical properties of the PEG and 1-TD composites. The PCM composites were connected with a thermoelectric power generator for energy harvesting. The increase in GNP portion in the PCM composites could achieve a high thermoelectric energy conversion efficiency and the 1:2 ratio PCM composites exhibited a more enhanced energy harvesting than other PCM composites. In addition, the collected electrical energy turned on the LED bulb successfully. The finite element method (FEM) was employed to calculate the temperature profiles. The numerical results agree with the experimental results. The optimal energy harvesting system was constructed by the combination of the PEG and 1-TD composites. These PCM composites were able to store and release a large amount of heat without any leakage to achieve renewable thermoelectric energy harvesting. The LED results show that the 1:2 PEG and 1:1 1-TD composite-based energy harvesting device induced the highest electrical current during heating and cooling. It is anticipated that the optimum energy harvesting system can be employed for a thermal sensor, a heat recovery device and a functional power generator.

## Figures and Tables

**Figure 1 nanomaterials-11-02192-f001:**
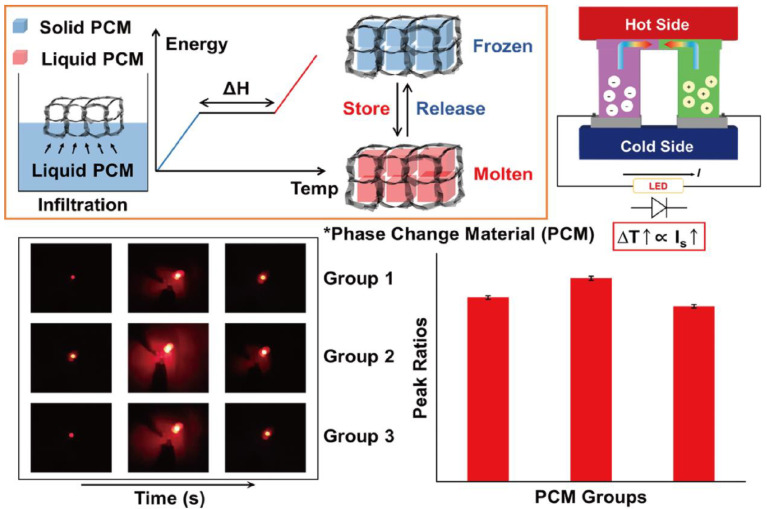
Schematics of the fabrication process for phase change material (PCM) composites by using the infiltration method. The thermoelectric energy harvesting system could generate induced current and turn on the LED bulb.

**Figure 2 nanomaterials-11-02192-f002:**
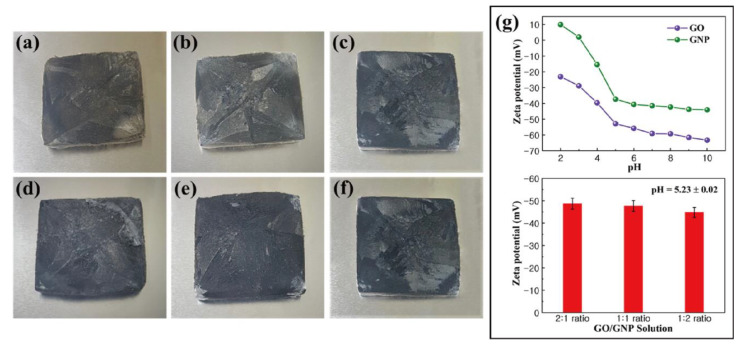
Photographs of graphene oxide (GO) aerogel and graphene nano-platelet (GNP): (**a**) 2:1 ratio GO/GNP aerogel, (**b**) 1:1 ratio GO/GNP aerogel, (**c**) 1:2 ratio GO/GNP aerogel, (**d**) 2:1 ratio rGO/GNP aerogel, (**e**) 1:1 ratio rGO/GNP aerogel and (**f**) 1:2 ratio rGO/GNP aerogel. (**g**) Zeta potential values of GO and GNP in the range of 2–10 pH.

**Figure 3 nanomaterials-11-02192-f003:**
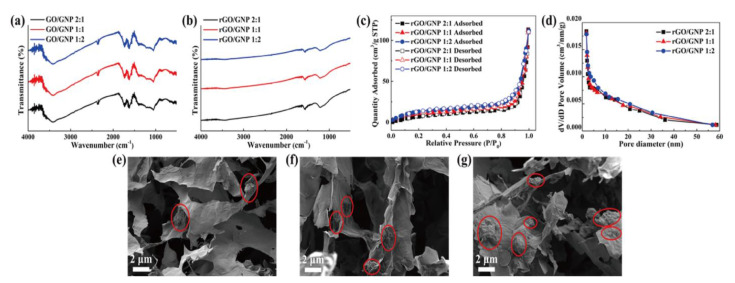
FT-IR peaks of (**a**) GO with different GNP ratios and (**b**) rGO/GNP. (**c**) Graphene aerogels BET peaks and (**d**) pore size distribution of the samples. SEM images of (**e**) 2:1 ratio rGO/GNP aerogel, (**f**) 1:1 ratio rGO/GNP aerogel and (**g**) 1:2 rGO/GNP aerogel.

**Figure 4 nanomaterials-11-02192-f004:**
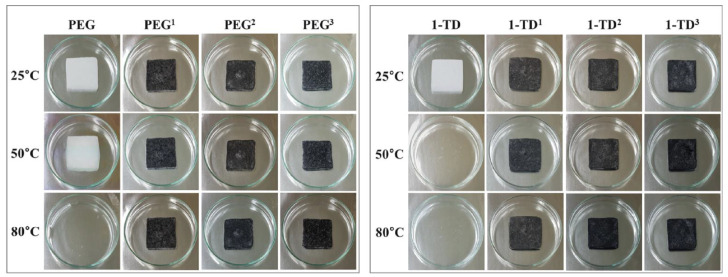
Photographs for the PCMs form-stable test. PEG1 has a 2:1 ratio of rGO/GNP, PEG2 has a 1:1 ratio and PEG3 has a 1:2 ratio. 1-TD composites are labeled in the same way.

**Figure 5 nanomaterials-11-02192-f005:**
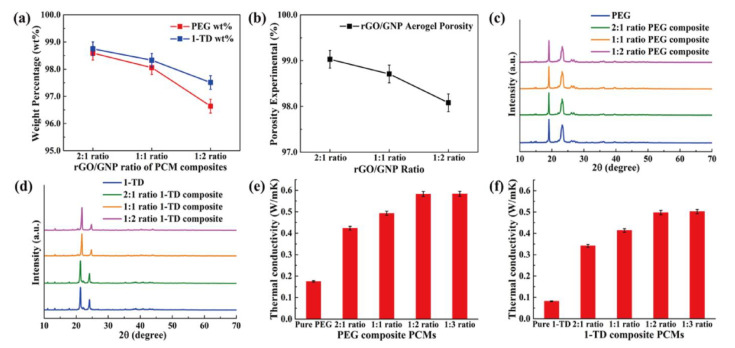
(**a**) Weight percentage of PEG and 1-TD in the PCM composite and (**b**) porosity of rGO/GNP aerogels. XRD peaks of (**c**) pure PEG and PEG composites and (**d**) pure 1-TD and 1-TD composites. Thermal conductivities of (**e**) pure PEG and PEG composites and (**f**) pure 1-TD and 1-TD composites.

**Figure 6 nanomaterials-11-02192-f006:**
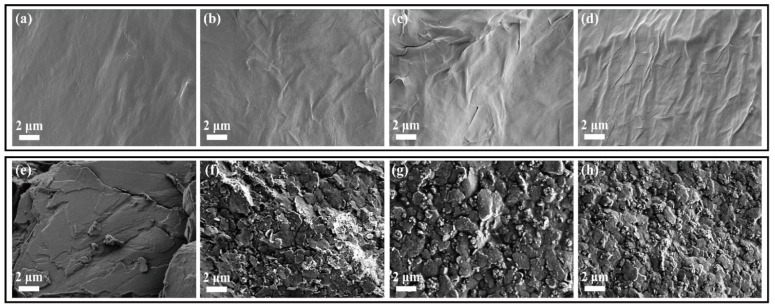
SEM images of (**a**) pure PEG, (**b**) 2:1 PEG composite, (**c**) 1:1 PEG composite, (**d**) 1:2 PEG composite, (**e**) pure 1-TD, (**f**) 2:1 1-TD composite, (**g**) 1:1 1-TD composite and (**h**) 1:2 1-TD composite.

**Figure 7 nanomaterials-11-02192-f007:**
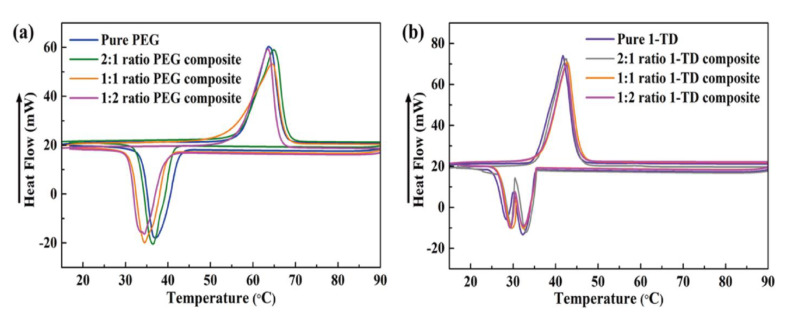
DSC curves of (**a**) pure PEG and PEG composites and (**b**) pure 1-TD and 1-TD composites.

**Figure 8 nanomaterials-11-02192-f008:**
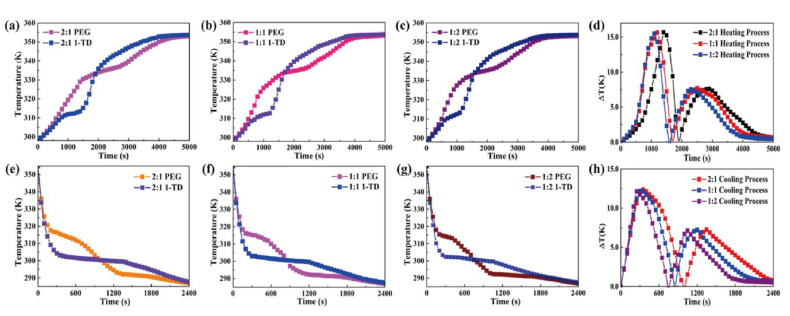
Temperature profiles during heating for (**a**) 2:1 ratio PEG and 1-TD composites, (**b**) 1:1 ratio PEG and 1-TD composites and (**c**) 1:2 ratio PEG and 1-TD composites. (**d**) Resulting temperature differences. Cooling cycle of (**e**) 2:1 ratio PEG and 1-TD composites, (**f**) 1:1 ratio PEG and 1-TD composites and (**g**) 1:2 ratio PEG and 1-TD composites. (**h**) Resulting temperature differences.

**Figure 9 nanomaterials-11-02192-f009:**
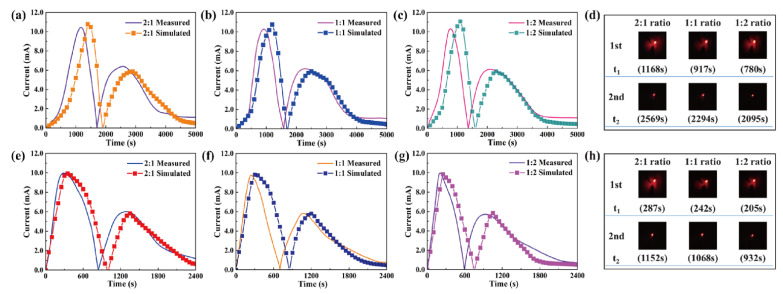
Experimental results and numerical calculations during heating for (**a**) 2:1 ratio PCM composites, (**b**) 1:1 ratio PCM composites and (**c**) 1:2 ratio PCM composites. (**d**) LED bulb images upon heating. Cooling results of (**e**) 2:1 ratio PCM composites, (**f**) 1:1 ratio PCM composites and (**g**) 1:2 ratio PCM composites. (**h**) LED bulb images upon cooling.

**Figure 10 nanomaterials-11-02192-f010:**
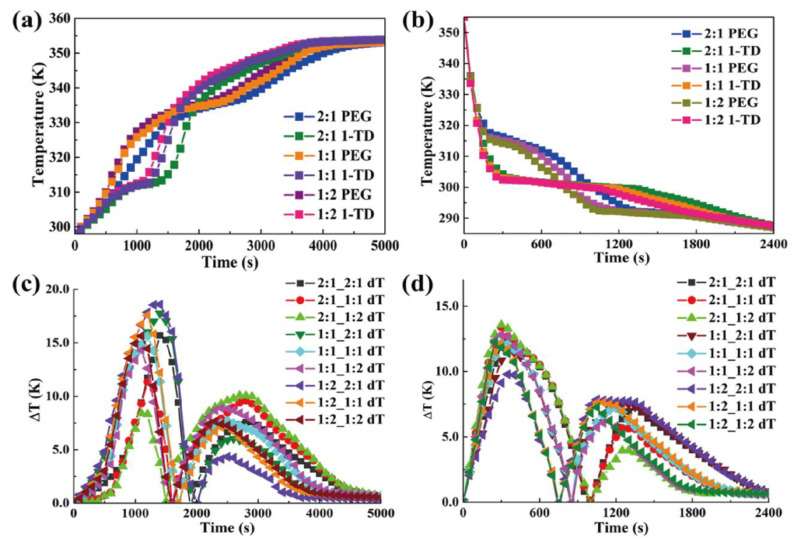
Temperature profiles for PCM composites with different GNP ratios during the (**a**) heating process and (**b**) cooling process. Temperature difference for PCM composite during the (**c**) heating process and (**d**) cooling process.

**Figure 11 nanomaterials-11-02192-f011:**
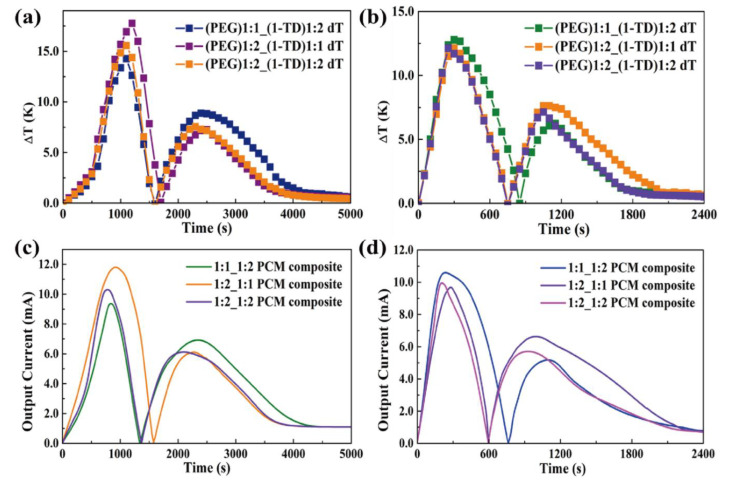
Optimization of components and temperature difference during the (**a**) heating process and (**b**) cooling process. Current curves of PCM composites during the (**c**) heating process and (**d**) cooling process.

**Figure 12 nanomaterials-11-02192-f012:**
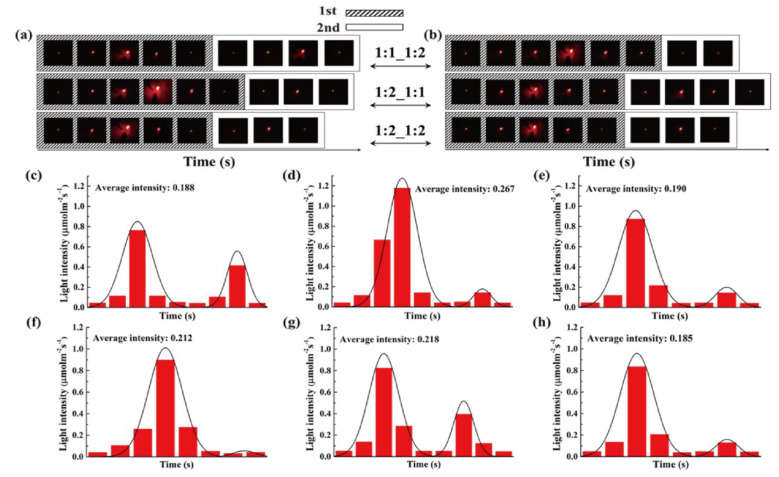
LED bulb images during the (**a**) heating process and (**b**) cooling process. Light intensities during heating for (**c**) 1:1 PEG and 1:2 1-TD composites, (**d**) 1:2 PEG and 1:1 1-TD composites and (**e**) 1:2 PEG and 1:2 1-TD composites. Light intensities during cooling for (**f**) 1:1 PEG and 1:2 1-TD composites, (**g**) 1:2 PEG and 1:1 1-TD composites and (**h**) 1:2 PEG and 1:2 1-TD composites.

**Figure 13 nanomaterials-11-02192-f013:**
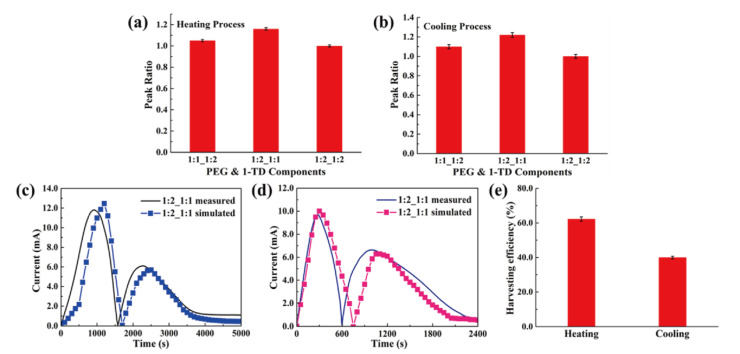
Peak ratios of PCM composites during the (**a**) heating process and (**b**) cooling process. Experimental current and numerical calculation during the (**c**) heating process and (**d**) cooling process. (**e**) Thermoelectric energy harvesting efficiency.

**Table 1 nanomaterials-11-02192-t001:** BET results of rGO/GNP aerogels with different ratios.

Samples	Graphene/GNP 2:1	Graphene/GNP 1:1	Graphene/GNP 1:2
Surface Area (m^2^/g)	373.95 ± 0.10	372.74 ± 0.10	370.36 ± 0.10

**Table 2 nanomaterials-11-02192-t002:** Weight percentages of PCM with different GNP mass ratios.

Samples	2:1 Ratio PCM Composite	1:1 Ratio PCM Composite	1:2 Ratio PCM Composite
PEG wt%	98.59 ± 0.10	98.06 ± 0.10	96.64 ± 0.10
1-TD wt%	98.75 ± 0.10	98.33 ± 0.10	97.51 ± 0.10

**Table 3 nanomaterials-11-02192-t003:** Porosities of rGO/GNP aerogels with different GNP mass ratios.

Samples	Graphene/GNP 2:1	Graphene/GNP 1:1	Graphene/GNP 1:2
Porosity (%)	99.03 ± 0.02	98.71 ± 0.02	98.08 ± 0.02

**Table 4 nanomaterials-11-02192-t004:** DSC results of pure PEG and 1-TD composites.

Samples	T_o_ (°C)		T_e_ (°C)		T_p_ (°C)		ΔH (J/g)	
	Heating Cycle	Cooling Cycle	Heating Cycle	Cooling Cycle	Heating Cycle	Cooling Cycle	Heating Cycle	Cooling Cycle
Pure PEG	50.75	41.54	69.16	32.68	66.81	37.50	182.62	164.84
2:1 PEG Composite	50.42	42.38	68.55	33.05	67.43	36.83	180.72	163.22
1:1 PEG Composite	47.64	42.51	68.31	33.17	67.41	34.59	180.56	163.03
1:2 PEG Composite	47.11	42.86	67.34	33.61	66.25	34.49	180.17	162.67
Pure 1-TD	34.47	36.11	45.23	20.23	41.63	27.68	226.09	213.82
2:1 1-TD Composite	34.36	36.14	45.18	20.31	42.27	26.74	221.87	210.13
1:1 1-TD Composite	34.18	36.22	45.15	20.56	42.51	25.32	220.76	209.79
1:2 1-TD Composite	34.03	36.35	44.97	20.64	42.24	25.01	220.43	208.56

**Table 5 nanomaterials-11-02192-t005:** Results of LED bulb during heating.

Time (s)	Onset (1st)	t_1_ (Max)	Endset (1st)	Onset (2nd)	t_2_ (Max)	Endset (2nd)	Total
1:1 PEG and 1:2 1-TD	544	838	1121	1778	2343	2968	1767
1:2 PEG and 1:1 1-TD	380	919	1422	1958	2274	2647	1731
1:2 PEG and 1:2 1-TD	472	779	1149	1746	2097	2691	1622

**Table 6 nanomaterials-11-02192-t006:** Results of LED bulb during cooling.

Time (s)	Onset (1st)	t_1_ (Max)	Endset (1st)	Onset (2nd)	t_2_ (Max)	Endset (2nd)	Total
1:1 PEG and 1:2 1-TD	84	235	627	1048	1099	1141	636
1:2 PEG and 1:1 1-TD	109	282	503	767	995	1404	1031
1:2 PEG and 1:2 1-TD	98	206	466	790	928	1108	686

## Data Availability

Data will be made available upon reasonable request to the corresponding author.

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
