# Peer review of "Analysis of Thermoelectric Energy Harvesting with Graphene Aerogel-Supported Form-Stable Phase Change Materials"

_nanomaterials, 2021, doi:10.3390/nano11092192_

Round 1

Reviewer 1 Report

Comments on Manuscript ID: nanomaterials-1338522

In this manuscript, the author constructed an advanced composite energy harvesting system by embedding GNP into PCM, which improved the thermal conductivity and mechanical properties of polyethylene glycol and 1-TD composites, and has a greater use of new energy in the future, and shows certain scientific interest and application value. However, there are still the following major issues that need to be corrected before published in Nanomaterials.

  1. There are some grammatical errors in this paper, please check and correct them carefully.
  2. There should be many types of phase change materials, not just PEG and 1-TD materials. In the preface, the author should be doing a brief review.
  3. In the experimental characterization section, "The PCM samples were heated from 15°C to 90°C at a speed of 10°C/min". However, in the chapter "4.2 The characterization of PCM composites", the author mentioned "All of sample was placed on the hot plate under the 25 °C and increased to 80°C". Why?
  4. Although both graphene and graphite sheet have excellent thermal conductivity, what is the difference between the thermal conductivity of the two? Perhaps rGO and GNP supported PEG-1TD composites should be supplemented.
  5. In Fig. 2g, the Zeta potential value of GO/GNP with different ratios decreases with the increase of GNP content. Why?
  6. In Fig. 5e, f, the thermal conductivity increases with the increase of GNP content. What is the difference between the thermal conductivity of GO and GNP?
  7. Please explain the meaning of the two characteristic peaks in the DSC curve in Fig. 7b.
  8. The references are mainly concentrated around 2020, and there are few in other years, please add some appropriately.

Author Response

We thanks to the referee for the vital comments. We finished all of comment and uploaded the response file for further review.

Reviewer 2 Report

The Authors presented an extensive work and obtained practically important results. Their manuscript brings together several important stages - the synthesis of graphene-based composites with phase-changing materials; the implementation of thermoelectric energy conversion; and theoretical calculations of thermal conductivity. The only drawback is the formula (5) for the thermal conductivity of a mixture of two phases. A similar formula is known for a mixture of thermodynamically similar materials (for example, two liquids). In the case of a mixture of liquid and solid phases, the formula (5) should be supported by an appropriate reference. In my view, the paper should be adopted after a minor revision.

Author Response

We thanks to the referee for the helpful suggestion. We finished the comment and uploaded the response file for further review.

Round 2

Reviewer 1 Report

This revised manuscript can be accepted.